# Fatty Acid Synthesis and Degradation Interplay to Regulate the Oxidative Stress in Cancer Cells

**DOI:** 10.3390/ijms20061348

**Published:** 2019-03-18

**Authors:** Valeryia Mikalayeva, Ieva Ceslevičienė, Ieva Sarapinienė, Vaidotas Žvikas, Vytenis Arvydas Skeberdis, Valdas Jakštas, Sergio Bordel

**Affiliations:** 1Institute of Cardiology, Lithuanian University of Health Sciences, LT 44307 Kaunas, Lithuania; Valeryia.Mikalayeva@lsmuni.lt (V.M.); Ieva.Cesleviciene@lsmuni.lt (I.C.); Ieva.Sarapiniene@lsmuni.lt (I.S.); Arvydas.Skeberdis@lsmuni.lt (V.A.S.); 2Institute of Pharmaceutical Technologies, Lithuanian University of Health Sciences, LT 44307 Kaunas, Lithuania; Vaidotas.Zvikas@lsmuni.lt (V.Ž.); Valdas.Jakstas@lsmuni.lt (V.J.)

**Keywords:** cancer, metabolism, lipids, oxidative stress, metabolic flux analysis

## Abstract

Both cytosolic fatty acid synthesis (FAS) and mitochondrial fatty acid oxidation (FAO) have been shown to play a role in the survival and proliferation of cancer cells. This study aimed to confirm experimentally whether FAS and FAO coexist in breast cancer cells (BCC). By feeding cells with ^13^C-labeled glutamine and measuring labeling patterns of TCA intermediates, it was possible to show that part of the cytosolic acetyl-CoA used in lipid synthesis is also fed back into the mitochondrion via fatty acid degradation. This results in the transfer of reductive potential from the cytosol (in the form of NADPH) to the mitochondrion (in the form of NADH and FADH_2_). The hypothesized mechanism was further confirmed by blocking FAS and FAO with siRNAs. Exposure to staurosporine (which induces ROS production) resulted in the disruption of simultaneous FAS and FAO, which could be explained by NADPH depletion.

## 1. Introduction

It has been known for more than 50 years that neoplastic tissues rely on de novo fatty acid synthesis (FAS) to proliferate [1]. The fatty acid synthase gene (*FASN*) was found to be a prognostic marker in breast [2] and prostate cancer [3]. ATP-citrate lyase, which is responsible for the synthesis of cytosolic acetyl-CoA in mammalian cells and fuels fatty acid synthesis, has also been shown to be necessary for cell transformation and tumor formation [4]. Acetyl-CoA carboxylase (ACC), which catalyzes the transformation of acetyl-CoA into malonyl-CoA and is necessary for fatty acid biosynthesis, was shown to induce growth arrest when inhibited chemically [5]. Overexpression of the oncogenic *Ras* has been associated with increased levels of long fatty acid chains [6] and non-small cell lung cancer tissues have also shown longer fatty acid chains [7], which suggests a higher activity of fatty acid elongases. A gene expression meta-analysis, in which genes were clustered forming metabolic sub-networks [8], revealed that the expression of metabolic sub-networks related to fatty acid synthesis and elongation, positively correlated with cell proliferation rates (using data from the NCI-60 collection) and negatively correlated with the survival prognosis of colon cancer patients. Interestingly, the same was observed for metabolic sub-networks related to fatty acid degradation and β-oxidation. These observations led to the hypothesis of both phenomena coexisting in the same cells.

The role of fatty acid oxidation (FAO) in the survival and proliferation of cancer cells is attracting growing attention [9,10]. FAO was found to inhibit apoptosis of leukemia cells [11]. *PPAR*-mediated fatty acid oxidation has been shown to protect cancer cells against apoptosis induced by loss of attachment to the extracellular matrix [12]. Carnitine palmitoyltransferase 1C (*CPT1C*), an element of the carnitine shuttle involved in the transport of cytosolic fatty acids to the mitochondrion (where FAO takes place), has also been identified as an oncogene [13].

Given all the previously mentioned findings, Carracedo et al. [9] stated that “a big challenge is to unify the idea of FAO as an essential pathway in cancer cells with the fact that cancer cells also require active FAS in order to grow and divide.” Simultaneous cytosolic FAS and mitochondrial β-oxidation has been considered impossible due to the inhibition of CPTI proteins by malonyl-CoA. CPTI proteins are responsible for the transfer of fatty acids from the cytosol to the mitochondrion via the so-called carnitine shuttle, while malonyl-CoA is an intermediate of FAS. However, malonyl-CoA can be produced by two enzymes, acetyl-CoA carboxylases 1 and 2 (ACC1 and ACC2), and genetic evidence suggests that while malonyl-CoA generated by ACC2 inhibits CPTI and therefore blocks fatty acid transport to the mitochondrion and its subsequent β-oxidation, malonyl-CoA generated by ACC1 does not exert suppressive effects on CPTI [14]. This is probably due to the direct channeling of malonyl-CoA from ACC1 to FASN, without being freely released in the cytosol.

Two main interpretations of the function of FAO in cancer cells are possible. First of all, it could be assumed that FAO plays a protective role only under conditions of metabolic stress such as loss of attachment to the extracellular matrix, during which glucose uptake and catabolism are suppressed [9]. Under such conditions, FAO would work as an alternative source of ATP [15] or NADPH [16], and FAS would have negative effects on cell survival as it would increase ATP and NADPH consumption [16]. Alternatively, FAO and FAS have been suggested to occur simultaneously and support each other [17]. This hypothesis is also supported by the fact that treatment of cells with orlistat (an inhibitor of lipid synthesis) resulted in decreased oxygen consumption rates [11], which was interpreted as the existence of simultaneous lipid synthesis and oxidation in the cells. Further evidence for a coexistence of FAS and FAO was provided by the fact that simultaneous targeting of FAS and FAO resulted in enhanced therapeutic effects in prostate cancer [18] and myeloma [19]. However, the available evidence is still indirect.

In this work we aimed to directly test the existence of simultaneous FAS and FAO by performing metabolic flux analysis using ^13^C-labeled glutamine in combination with gene silencing using small interfering RNAs (siRNAs). Second, we examined the relevance of such a cycle for the resistance of breast cancer cells to oxidative stress.

## 2. Results

### 2.1. Assessment of Simultaneous FAS and FAO Using ^13^C Labeling

Fatty acids are synthesized from cytosolic acetyl-CoA, which itself is partially derived from citric acid. Citric acid is originated in the tricarboxylic acid (TCA) cycle and transported to the cytosol, but can also be originated from glutamine via the reductive carboxylation of α-ketoglutarate [20]. FAO takes place in the mitochondrion (fatty acids are transported from the cytosol via the carnitine shuttle), resulting in mitochondrial acetyl-CoA, which feeds the TCA cycle. The simultaneous occurrence of both processes implies that part of the acetyl-CoA feeding the TCA cycle is derived from citrate. Supplying ^13^C-labeled glutamine will result, via reductive carboxylation, in the production of citrate labeled in five carbons (M5 isotopomer) (Figure 1a), which itself will give rise to labeled cytosolic acetyl-CoA taking part in the synthesis of lipids. If these lipids are being simultaneously degraded, labeled acetyl-CoA will be entering the TCA cycle, resulting in the formation of fully labeled citric acid molecules (M6 isotopomer) (Figure 1a). 

Alternatively, the presence of fully labeled citrate could be explained by the formation of labeled pyruvate obtained from malate via the malic enzyme (ME). Therefore, in order to assess if lipid metabolism is actually playing a role in the formation of fully labeled citrate, we combined labeling experiments with gene silencing using siRNAs. Previously described BCCs [21] were selected as a candidate for testing the existence of simultaneous FAS and FAO, as other two cell lines MCF7 and BT-474 showed notably lower amount of fully labeled citrate. The percentage of M6 citrate in MCF7 and BT-474 was 1.15% and 0.5%, respectively (Figure 1b).

BCC cells were alternatively transfected with siRNAs against the *FASN* and the *ECHS1* genes, which are involved in FAS and FAO, respectively, and compared to a mock transfection (see Methods). The cells were also transfected with siRNAs against the malic enzyme (*ME*1-3) genes, in order to assess the importance of the malic enzyme for the formation of fully labeled citric acid (western blot analysis is represented in Appendix A. The siRNAs against *FASN* and *ECHS1* resulted in a very similar drop of the M6 fraction (Figure 1c), which was in both cases statistically significant (from 5.8 ± 0.3% to, respectively, 4.2 ± 0.2%.; n = 4; *p* = 0.0054 and 4.2 ± 0.3%; n = 4; *p* = 0.0065). This confirms that part of the fully labeled citrate appears to be generated via the hypothesized mechanism of simultaneous FAS and FAO. An even stronger drop (to 2.4 ± 0.3%; n = 4, *p* = 0.0001) was caused by siRNAs targeting ME (Figure 1c), which suggests that both mechanisms of formation of fully labeled citrate are active.

### 2.2. Assessment of Simultaneous FAS and FAO Using Measurements of Mitochondrial Membrane Potential

FAS occurs in the cytosol and requires the oxidation of two NADPH molecules per acetyl-CoA. On the other hand, mitochondrial FAO results in the generation of one NADH and one FADH_2_ molecules per acetyl-CoA, both of which can be used to feed the respiratory chain (Figure 1a). This results in the transfer of redox potential from the cytosol to the mitochondrion, which can be used to maintain higher mitochondrial membrane potentials. ME catalyzes a reaction producing NADPH, which cannot be used in the respiratory chain. Therefore, if simultaneous FAS and FAO is taking place, it can be expected that silencing of FASN or ECHS1 will result in a drop of the mitochondrial membrane potential, while silencing of ME is not expected to have any significant impact upon it. Mitochondrial potential was estimated by flow cytometry using JC-1 dye (see Methods). Cells transfected with siRNAs targeting FASN and ECHS1 showed significant drops in mitochondrial membrane potential (from 10.9 ± 0.7 to, respectively, 6.9 ± 0.7; n =3; *p* = 0.014 and 7.5 ± 0.1; n = 3; *p* = 0.009; Figure 1d). Cells transfected with siRNAs targeting ME did not show a significant drop in membrane potential (9.2 ± 0.8; *p* = 0.19). 

### 2.3. Metabolic Flux Analysis of BCC Cells

Experimental labeling patterns of citrate and malate were used to adjust the metabolic flux distributions in the central carbon metabolism. The flux distributions were calculated using a model that contained the TCA cycle, FAS, FAO, glutaminolysis, reductive carboxylation of α-ketoglutarate, malic enzyme, pyruvate carboxylase, and pyruvate dehydrogenase. The mitochondrial and cytosolic pools of citrate and α-ketoglutarate were treated as single pools, as if these compounds were freely transferred across the mitochondrial membrane. The cytosolic oxaloacetate produced by citrate lyase was assumed to be transferred back to the mitochondrion.

The EMU (elementary metabolic unit) framework [22] was used to model isotopic distributions. The fitting (based on minimizing relative errors between the experimental and predicted labeling patterns) was performed as described in Materials and Methods. The flux distributions are expressed relatively to the rate of pyruvate dehydrogenase. 

Fitted metabolic flux distributions are represented in Figure 2a for the reference experiment and for the experiments with siRNAs (FASN, ECHS1, and ME). The data fitting revealed simultaneous FAS and FAO rates equal to 0.69 times the rate of pyruvate dehydrogenase and a flux of 0.48 in the ME reaction (for the control experiment). In the cells transfected with siRNAs against FASN or ECHS1, the solution minimizing the relative errors in isotopomer distributions (see Methods) corresponded to zero flux in the FAO reaction. Simultaneous transfection with siRNAs targeting three ME genes led to an estimated flux in the ME reaction of 0.03. 

The experimental and fitted isotopomer distributions for malate and citrate in “control/reference” experimental conditions and after siRNA silencing are shown in Figure 2b–i, respectively. Reduced fractions of M6 citrate isotopomers indicate reduced FAS, FAO, or ME activity. The isotopomer distributions of malic acid are mainly determined by the rate of glutaminolysis and are not expected to be modified by changes in FAS, FAO, and ME activities.

### 2.4. Effects of Oxidative Stress on Metabolic Fluxes

The overall effect of the described metabolic cycle formed by FAS and FAO is the shuttling of redox equivalents from the cytosol (in form of NADPH) to the mitochondrion (in form of NADH and FADH_2_). Cytosolic NADPH has two main physiological roles: the biosynthesis of biomass components and the resistance to oxidative stress (the last one is coupled to glutathione reductase and glutathione peroxidase). Increased production of reactive oxygen species (ROS) is therefore expected to result in the depletion of cytosolic NADPH, and subsequently, to stop FAS and other processes that require NADPH. BCC cells treated with staurosporine, an alkaloid that induces high ROS levels [23], caused a strong drop of fully labeled citric acid (Figure 3), which is consistent with the expected arrest of FAS (a full arrest according to the metabolic flux analysis). This result suggests that BCC cells under normal conditions produce more cytosolic NADPH than what is necessary for their biosynthetic needs. In conditions of low oxidative stress, the excess redox potential is channeled into the mitochondrion, where it is ultimately used to produce ATP, while under conditions of high oxidative stress, the excess of NADPH can be directly used to eliminate cytosolic ROS.

### 2.5. Resistance of BCC and BT-474 Cells to Oxidative Stress

The previous results indicate that cells with simultaneous FAS and FAO have high cytosolic NADPH production rates, which confers to them the ability to counteract increases in oxidative stress. In contrast, we could expect cells not showing simultaneous FAS and FAO to have lower cytosolic NADPH production rates and also lower tolerance to increased oxidative stress. In order to test the mentioned hypothesis, we compared the effects of staurosporine on BCC and BT-474 cells. The choice of BT-474 cells for the comparison is due to the fact that they showed 10 times less M6 citrate than BCCs (Figure 1), which suggests that the loop formed by simultaneous FAS and FAO is not active in this cell line (Figure 4a). The ^13^C-labeling data revealed a zero flux in FAO rate for BT-474 cells.

Staurosporine treatment induced stronger mitochondrial depolarization in BT-474 cells compared to BCC, which was determined by the decrease of R/G ratio 3.4 ± 0.4 and 1.8 ± 0.05-fold (*p* < 0.001), respectively (Figure 4d,e). These results are consistent with our hypothesis of BCC cells being more resistant to oxidative stress.

## 3. Discussion

FAO has been recently shown to play a key role on the survival of cancer cells. On the other hand, FAS is also necessary for cancer cell growth and proliferation. There are two possible solutions in this apparently contradictory situation. It is possible that FAO helps cell survival in situations of metabolic stress when other sources of energy and redox cofactors are not available; on the other hand, even if FAS and FAO have traditionally been considered incompatible due to the inhibitory effects of malonyl-CoA on the carnitine shuttle (responsible for the transport of fatty acids into the mitochondrion), growing evidence is supporting the hypothesis of both phenomena coexisting and feeding each other in some cancer cells. In this work, we demonstrate that this is the case in BCC cells. The coexistence of FAS and FAO is the best explanation for the impact that siRNAs targeting the FASN and ECHS1 genes had on the amount of fully labelled citrate (when ^13^C-labelled glutamine is supplied to the cells) and the mitochondrial membrane potential. Blocking of FAS resulted in a very similar drop of fully labeled citrate and mitochondrial potential than blocking FAO. This is also consistent with previous observations [11] in which blocking lipid synthesis resulted in lower oxygen consumption.

A possible physiological role of this simultaneous occurrence of FAS and FAO is the shuttling of reductive potential contained in cytosolic NADPH to the mitochondrion. This would allow the cells to recycle the excess NADPH produced in the cytosol and channel its reductive potential into the respiratory chain. If cells are challenged by sudden increases of oxidative stress, this excess production of cytosolic NADPH could be immediately diverted to eliminate ROS, stopping lipid synthesis. This phenomenon indeed appears to take place in BCCs treated with staurosporine, in which a strong drop in the fraction of fully labelled citrate was observed compared to the untreated cells.

Metabolic cycles involving the oxidation of overproduced NADPH have previously been observed in microorganisms [24], where this apparent waste of energy could be compensated for by an increased robustness against sudden increases of oxidative stress. The case presented here suggests that human cells could also use analogous mechanisms.

Cells not showing this cooperative operation of FAS and FAO such as BT-474 are therefore likely to have lower production rates of cytosolic NADPH and to be more sensitive to sudden ROS increases. This could be one of the reasons for the stronger effects of exposure to staurosporine observed in BT-474 compared to BCC.

The presented findings contribute to the understanding of the interplay between FAS, FAO, and NADPH metabolism in cancer cells and confirm FAO as a suitable anticancer metabolic target for some cancer types.

## 4. Methods

### 4.1. Cell Lines and Culture Medium

BT-474 (human breast ductal carcinoma cell line; ATCC HTB-20, Manassas, VA, USA), MCF7 (human breast adenocarcinoma cell line; CLS-Cell Lines Service, Eppelheim, Germany), and BCC cells [21] were grown in Dulbecco’s Modified Eagle Medium:Ham’s F-12 (1:1; DMEM/F-12) (Life technologies, Carlsbad, CA, USA) medium, supplemented with 10% fetal bovine serum (FBS), 100 U/mL penicillin and 100 µg/mL streptomycin. Cells were maintained at 37 °C in a humidified incubator with 5% CO_2_. For treatment with staurosporine (Calbiochem, Darmstadt, Germany), cells were incubated for 4 h with 0.25 μM of latter compound.

### 4.2. siRNA Transfection

We used a jetPRIME transfection reagent (Polyplus-transfection^®^ SA, France) and siRNAs for gene silencing according to the manufacturer’s instructions. Final concentrations of 10 nM ECHS1 siRNA (ATGATGTGTGATATCATCTAT; Qiagen, Germantown, MD, USA) or FASN siRNA (CAGGCTTCAGCTCAACGGGAA; Qiagen) were used. Cells were simultaneously transfected with three different siRNAs against the different malic enzyme (ME1, ME2, and ME3) genes: 4nM ME1 (UGCCAUGACUCAGCGUUCtt; Ambion, Waltham, MA, USA), 4 nM ME2 (GGGUGUCUAUGGAAUGGGAtt; Ambion), 4 nM ME3 (GCCUUUACCCUUGAAGAAAtt; Ambion). All following procedures were performed 24 h after siRNA transfection. Mock transfection was used as control.

### 4.3. ^13^C Labeling

Cells were incubated in DMEM without glucose, L-Glutamine and piruvate (Sigma, Schnelldorf, Germany), supplemented with 10% FBS and antibiotics. L-Glutamine-^13^C_5_ (Aldrich, Hamburg, Germany) was added to the final concentration of 4 mM. Cells were incubated for 24 h. Metabolite extraction was performed as described earlier [25].

### 4.4. UPLC-ESI-MS Conditions

Separation of organic acids in samples was carried out with an Acquity H-Class UPLC system (Waters, Milford, MA, USA) equipped with a YMC-Triart C18 (100 × 2.0 mm, 1.9 µm) column (YMC, Kyoto, Japan). Triple quadrupole tandem mass spectrometer Xevo TQD (Waters, Milford, MA, USA) with an electrospray ionization (ESI) source was used to obtain mass spectroscopy (MS) data. The column temperature was maintained at 40 °C. Gradient elution was performed with a mobile phase consisting of 0.1% formic acid water solution (solvent A) and acetonitrile (solvent B) with the flow rate set to 0.4 mL/min. The initial conditions were set to 95% of solvent A. Linear gradient profile was applied with following proportions of solvent A: 0 to 0.2 min was set to 95%, 0.2 to 1.5 min–10%, 1.5 min to 1.8 min was maintained at 90%, and 1.8 to 2 min set back to initial conditions. Total time of analysis with equilibration was 3 min. Negative electrospray ionization was applied for analysis with the following settings: capillary voltage was set to negative 2 kV, source temperature applied at 150 °C, desolvation temperature of supplied nitrogen gas was set to 400 °C, desolvation gas flow -700 L/h, cone gas flow -20 L/h. Cone voltage was set to 25 V. MS data was collected in full scan mode in the 50 *m/z* to 250 *m/z* range. MS spectra and chromatograms of citrate isotopomers are presented in the Appendix A.

### 4.5. Fitting of Metabolic Flux Distributions

Mass distributions were calculated using a Python function (available in the Appendix A). This function uses a simplified model of central carbon metabolism containing the reactions depicted in Figure 1. The function is an implementation of the EMU framework [22]. The model contained seven independent reaction rates. One of them (pyruvate dehydrogenase—PDH) was set to 1 and all the other fluxes were given relative to the flux in PDH. The remaining independent reactions were the FAO rate, the malic enzyme, the formation of α-ketoglutarate derived from glutaminolysis, the carboxylation of α-ketoglutarate, pyruvate carboxylase, and the import of pyruvate into the mitochondrion.

The objective function to be minimized was the squared sum of relative errors in the labeling patterns of malate and citrate:(1)$$F=\overset{4}{\underset{i=1}{\sum }}{\left(\frac{M_{i}^{mal}-{\left(M_{i}^{mal}\right)}_{exp}}{{\left(M_{i}^{mal}\right)}_{exp}}\right)}^{2}+\overset{6}{\underset{i=1}{\sum }}{\left(\frac{M_{i}^{cit}-{\left(M_{i}^{cit}\right)}_{exp}}{{\left(M_{i}^{cit}\right)}_{exp}}\right)}^{2}$$

The contribution of *M*_0_ was not included due to being redundant (the sum of all the mass fractions has to be equal to 1). Previous to the fitting, an identification analysis was performed by numerically calculating the Jacobian matrix of the system and performing its singular value decomposition. The value of the lowest singular value of the Jacobian matrix measures how far the columns of this matrix are from being linearly dependent. A lowest singular value equal to zero would mean that there are many different flux distributions leading to the same labeling patterns and the system is therefore not identifiable. In order to assess the contribution of each flux to the identifiability of the system, new Jacobian matrices were calculated by subtracting the column corresponding to the tested flux and performing singular value decomposition on the new matrix (see Appendix A). The higher that the new lowest singular value was, the more difficult to identify was the subtracted flux. The results of this analysis are presented in the Appendix A. FAO (the flux most interesting in this work) was the most difficult flux to identify. In order to circumvent this problem, we performed a global optimization for the experiments corresponding to the mock transfection and the silencing of *FASN* and *ECHS1*. This approach is based on the assumption that all the independent fluxes, except FAO are unaffected by the siRNAs. The carboxylation rate of α-ketoglutarate was also allowed to change between conditions, as this flux has been reported to be very sensitive to small perturbations in the concentrations of α-ketoglutarate and citric acid [20].

The fitting of the fluxes corresponding to ME silencing, was performed by fixing the rate of FAO to the value estimated in the previous fitting and allowing all the other fluxes to change. Fitting of flux distributions after treatment with staurosporine and of BT-474 cells were performed allowing all the independent reactions to change.

For each fitting, the values of all the fluxes were sequentially incremented or diminished by 0.001 (without allowing them to take negative values) and if this increment resulted on a reduction of the objective function, the new values were kept and the process reiterated. Different initial sets of flux distributions were used and the fitting leading to the lowest relative error was kept.

### 4.6. Western Blot Analysis

BCC cells (3 × 10^6^) were lysed in ice-cold cell extraction buffer (Invitrogen, Carlsbad, CA, USA) supplemented with 20 μL/mL protease inhibitor cocktail (Sigma-Aldrich, Germany) and 1 mM PMSF (Abcam, Cambridge, U.K.) for 30 min. The lysates were centrifuged at 13,000 rpm for 10 min at 4 °C. Qubit® protein assay kit (Invitrogen) was used to determine total protein concentration by Qubit 3.0 fluorometer (Invitrogen). Cell lysates (30 μg) were separated by Bolt™ 4–12% Bis-Tris plus gels (Invitrogen) in MES SDS running buffer and transferred to 0.45 μm PVDF membranes (GE Healthcare, U.K.). Proteins were detected using primary antibodies against ECHS1 (ab174312) and FASN (ab99359) (Abcam), PPAR-δ (PA1-823A), ME1 (MA5-23524), ME2 (PA5-38007), ME3 (PA5-36494), and GAPDH (AM4300) (Thermo Fisher Scientific, USA), and a WesternBreeze® chemiluminescent kit (Invitrogen) according to the manufacturer’s instructions. Bands were visualized by G:Box Chemi Gel Documentation system (Syngene, Frederick, MD, USA).

### 4.7. Flow Cytometry Assay

Initially, 5 × 10^4^ cells were seeded in a 35 mm dishes in 2 mL of full media. Mock-transfected and siRNA-transfected cells were incubated with media containing 1 mg/L JC-1 dye (Biotium, Fremont, CA USA) for 20 min at room temperature. Afterwards, cells were washed with PBS, trypsinized, and collected using centrifugation. Then cells were resuspended in PBS with 5 mg/L 7-AAD dye (Millipore, Burlington, MA, USA) and incubated for 10 min. For positive control 10 μM carbonyl cyanide m-chlorophenylhydrazone (CCCP) was used. Samples were quantified using a Guava PCA flow cytometer (Millipore). The data were analyzed by guavaSoft 2.7 InCyte software. Mitochondrial membrane potential was represented as the JC-1 aggregates (red fluorescence) to monomers (green fluorescence) ratio (R/G).

### 4.8. Statistical Analysis

Error bars correspond to standard errors (number of samples specified in the figure legends) when two experiments were compared to each other. When predicted isotopomer distributions were compared to calculated ones, error bars of the experimental values correspond to 95% confidence intervals. Comparisons between two values were performed using the Student *t*-test. Statistical analysis was performed using Excel and the library scipy.stats.

## Figures and Tables

**Figure 1 ijms-20-01348-f001:**
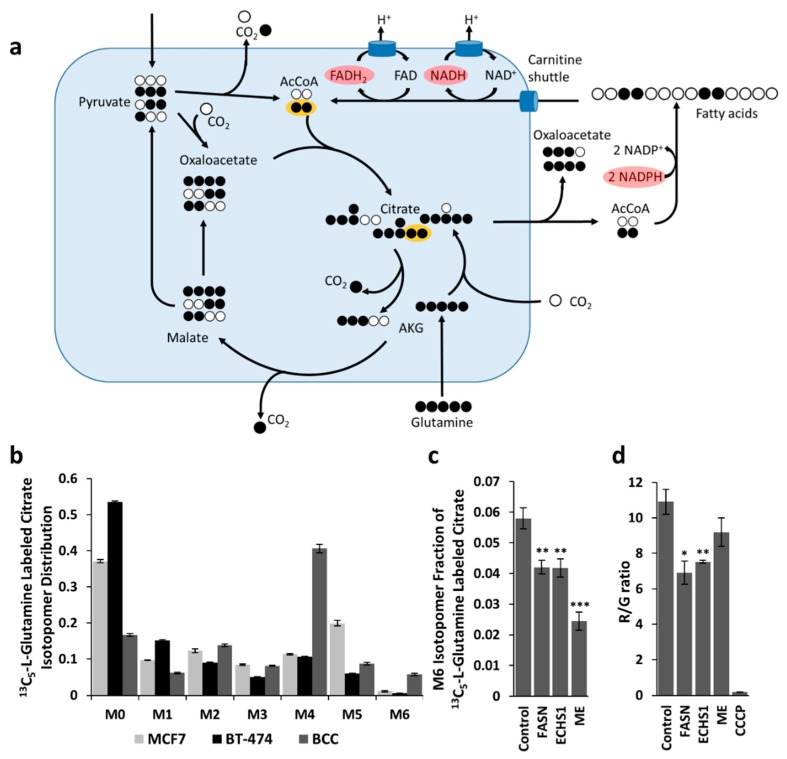
Experiments assessing the existence of simultaneous FAS and FAO. (**a**) Metabolic model with atomic transitions showing how degradation of fatty acids derived from glutamine or ME activity can result in the M6 citrate isotopomer, and scheme showing the shuttle of redox potential from cytosolic NADPH to the mitochondrion, formed by simultaneous FAS and FAO. Labeled carbon atoms entering the TCA cycle from FAO are indicated in yellow. (**b**) Percentages of fully labeled citrate in three different breast cancer cell lines which were fed with ^13^C_5_-L-Glutamine. BCC showed 5 and 10 times more fully labeled citrate than MCF7 and BT-474 cell lines, respectively, which makes it more likely to have simultaneous FAS and FAO. Bar graphs represent mean ± SEM (n = 4). (**c**) Experimental changes in the fraction of M6 citrate resulting from the silencing of FASN, ECHS1, and ME, respectively. Bar graphs represent mean ± SEM (n = 4). (**d**) Effects of silencing FASN, ECHS1, and ME on the mitochondrial membrane potential. Bar graphs represent mean ± SEM of aggregate/monomer (R/G) ratios of JC-1 dye (n = 3). * *p* < 0.05, ** *p* < 0.01, *** *p* < 0.001.

**Figure 2 ijms-20-01348-f002:**
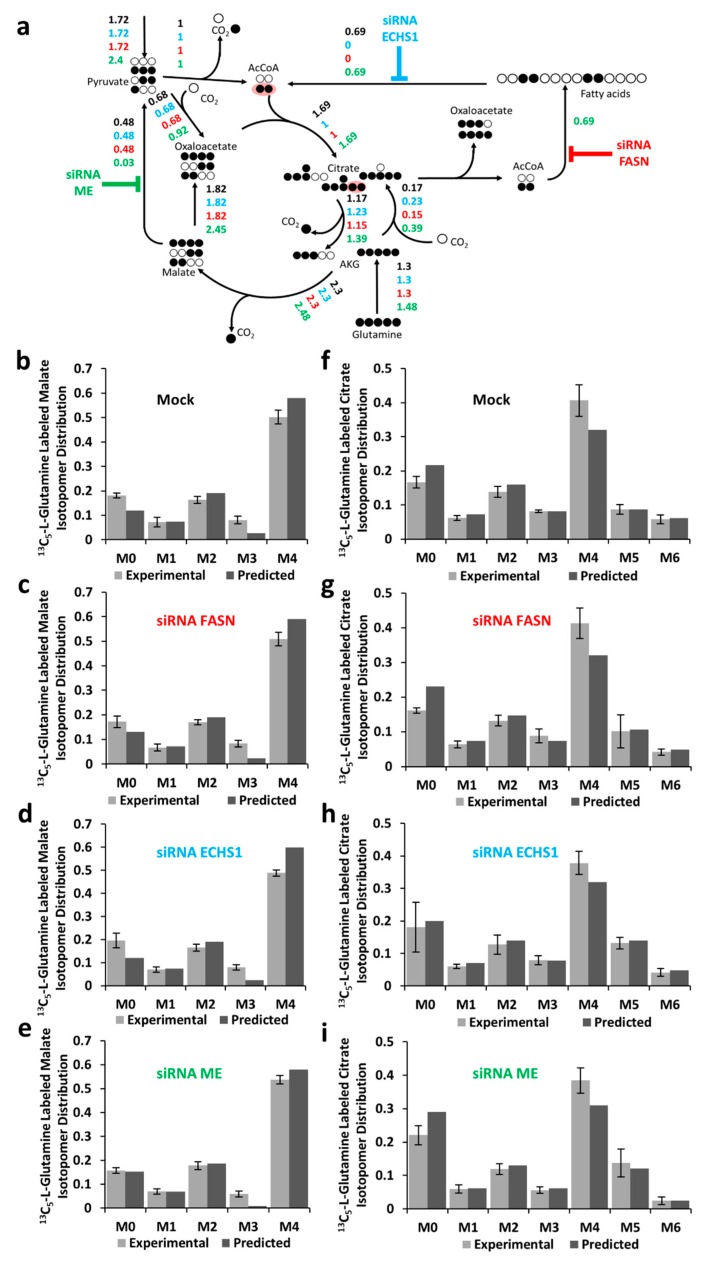
Estimated metabolic flux distributions and isotopomer distributions (simulated and experimental) of malate and citrate for BCC cells. (**a**) Metabolic model with atomic transitions and estimated flux distribution for the mock (black) and siRNAs against FASN (red), ECHS1 (cyan), and ME (green) transfected BCC cells. (**b**–**e**) Malate and (**f**–**i**) citrate isotopomer distribution in mock and siRNAs against FASN (red), ECHS1 (cyan), and ME (green) transfected BCC cells. The bar plots show the fractions of citrate and malate labeled in zero (M0) to six (M6) carbons. Error bars represent 95% confidence intervals (n = 4).

**Figure 3 ijms-20-01348-f003:**
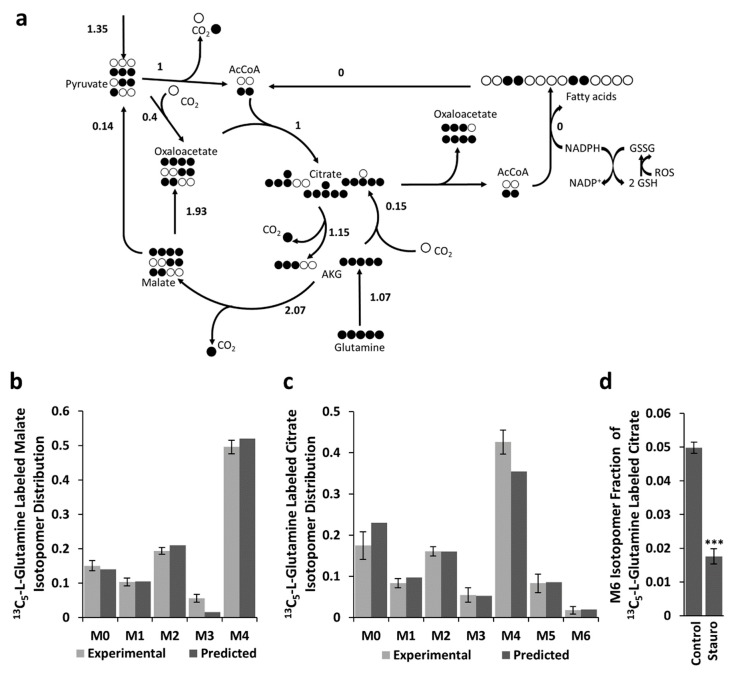
Estimated metabolic flux distributions and labeling patterns (simulated and experimentally observed) of malate and citrate for BCCs after treatment with staurosporine. (**a**) Estimated metabolic flux distributions for BCC cells treated with staurosporine. Simulated and observed labeling patterns of malate (**b**) and citrate (**c**) for BCCs treated with staurosporine. Error bars are 95% confidence intervals (n = 4). Drop of the fraction of M6 citrate caused by staurosporine (**d**). Bar graphs represent mean ± SEM (n = 4), *** *p* < 0.001.

**Figure 4 ijms-20-01348-f004:**
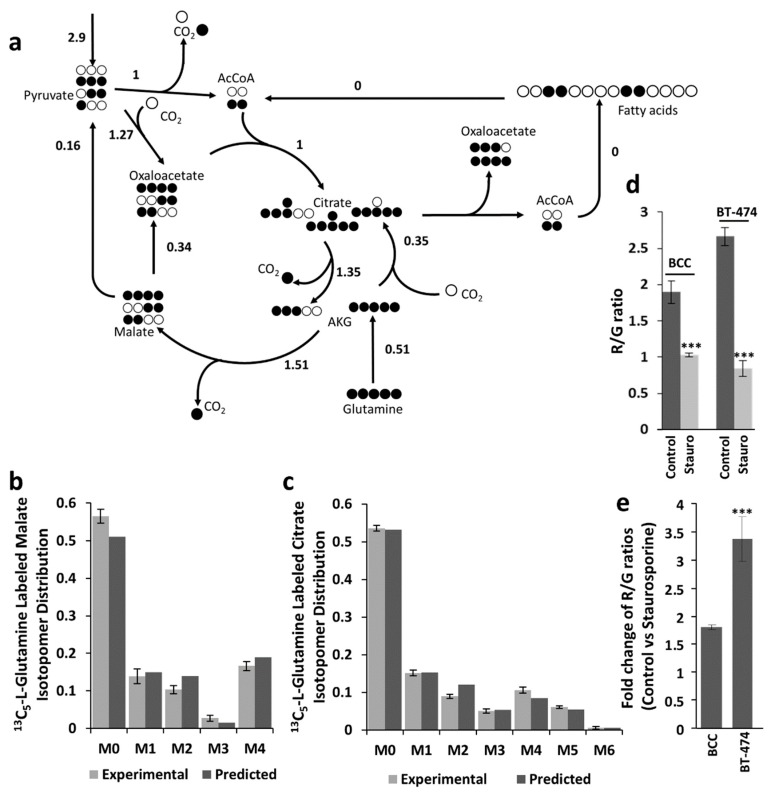
Estimated metabolic flux distributions and labeling patterns (simulated and experimental) of malate and citrate for BT-474 cells. (**a**) Estimated metabolic flux distributions for BT-474 cells. Simulated and observed labeling patterns of malate (**b**) and citrate (**c**). Error bars represent 95% confidence intervals (n = 4). (**d**) Staurosporine-induced mitochondrial depolarization in BT-474 and BCC cells. (**e**) Fold change of R/G ratios in BCC and BT-474 cells after staurosporine treatment. Bar graphs represent mean ± SEM (n = 4), *** *p* < 0.001.

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
