# Peer review of "Fatty Acid Synthesis and Degradation Interplay to Regulate the Oxidative Stress in Cancer Cells"

_ijms, 2019, doi:10.3390/ijms20061348_

Round 1
Reviewer 1 Report
In this manuscript, the authors address a novel and interesting topic regarding the roles of FAS and FAO in the specific case of breast cancer cell lines. Stable isotope labeling is used to clearly assess FAS and FAO concurrently and clear observations are presented to support the hypothesis of simultaneous FAS and FAO. Data supporting the subsequent conjectures on the role of FAS/FAO in the response to oxidative stress, however, could be more robust, for example by using more than one method of inducing oxidative stress as well as more indicators.
The authors should also take care to proof-read the language in the manuscript, especially in the abstract section.
Author Response
First of all we would like to thank the reviewers for their comments and the opportunity to improve the manuscript.
We repeat each comment below with an answer following it:
In this manuscript, the authors address a novel and interesting topic regarding the roles of FAS and FAO in the specific case of breast cancer cell lines. Stable isotope labeling is used to clearly assess FAS and FAO concurrently and clear observations are presented to support the hypothesis of simultaneous FAS and FAO. Data supporting the subsequent conjectures on the role of FAS/FAO in the response to oxidative stress, however, could be more robust, for example by using more than one method of inducing oxidative stress as well as more indicators.
We agree with the reviewer on the fact that alternative methods to induce oxidative stress could be used besides staurosporine and we are open to this possibility if it is considered necessary.
On the other hand, the hypothesis that we aimed to prove with the experiments involving staurosporine is the decrease of M6 citrate as a result of depletion of the cytosolic NADPH pool. NADPH will be depleted by any increase of ROS, as it is used to reduce oxidant species in a mechanism coupled to glutathione reductase and glutathione peroxidase. Staurosporine is commonly used as an inductor of apoptosis and it has been solidly established that staurosporine induces apoptosis by increasing ROS (Neurochem Int. 2009 55(7):581-92; Experimental Neurology 2001 171: 84-97; Journal of Biological Chemistry 2004 279, 50499-50504). Indeed, antioxidant agents, such as the superoxide dismutase EUK-134 (Journal of Biological Chemistry 2004 279, 50499-50504) have been found to attenuate the apoptotic effects of staurosporine.
Treatment with staurosporine decreased M6 citrate, which was consistent with our initial hypothesis. Even if staurosporine is reported to have other effects, neither alternative ways in which it could induce NADPH depletion nor possible direct inhibition mechanisms of fatty acid synthesis or beta oxidation, have been described to our knowledge.
Reviewer 2 Report
“Fatty acid synthesis and degradation interplay to regulate the oxidative stress in cancer cells” by Mikalayeva et al.
The manuscript deals with the challenging hypothesis that fatty acid synthesis and -oxidation occur simultaneously in (breast) cancer cells. The authors use metabolic flux studies with 13C labeled glutamine in combination with gene silencing. They use staurosporine as an inducer of mitochondrial depolarization to induce oxidative stress.
The role of fatty acid metabolism in cancer is topical and the authors present the current views and state of literature clearly in their introduction. The manuscript is well written and very understandable. I am not an expert on metabolic fluxes and cannot give a meaningful opinion on the mathematical approach of the study.
There are some issues that need to be resolved before a definitive decision can be made regarding the quality of the data and whether or not the conclusions drawn are supported by this data.
Major Points.
The abstract does not adequately cover the work. For instance, the experimental approach via flux studies and siRNA is not mentioned. The sentence ‘Its production in excess and its shuttling to the mitochondrion (using simultaneous FAS and FAO) could be a mechanism that allows the cell to respond immediately to sudden increases of oxidative stress.’ is a very big leap for the reader.
I have serious concerns regarding the UPLC/MS experiments. A triplequad delivers low resolution data and use a ‘scanning’ mode of operation, the number of spectra per second is low. The authors combine this ‘slow’ and low resolution MS instrument with very, very fast gradient LC (2 min). I expect a lot of coelution of nominally isobaric compounds. Currently, there is no way readers can assess data quality. I, therefore, suggest that the authors present overlays of extracted ion chromatograms of all isotopomers of compounds they identified. This way, the reader can be reassured that the chromatographic quality of the peaks, and the overlap of isotopomers) is sufficient to support the digested data and the conclusions drawn.
I cannot open the cdf files in the supplementary data, in an effort to assess the LCMS data quality. I have tried to import data to XCMS, mzMine, Toppview KNIME and SEEMS but all viewers produce errors. Please convert the data to something more readable such as mzML.
The use of staurosporine to induce ROS formation, seems unfortunate. Staurosporine is a very aspecific but potent inhibitor of a wide variety of protein kinases. Indeed, it also induces apoptosis which is linked to ROS production, but I find that the claim that staurosporine effects are due to ROS formation lacks evidence. I appreciate the measurement of mitochondrial depolarization, but this doesn’t preclude that the observed effects are due to other changes induced by staurosporine.
Minor points
- Some spelling issues. E.g. ‘piruvate’ (p9 line 14)
- The file mentioned in the supplementary material.doc is not present (‘mass spectra raw data have been uploaded in the compressed file raw_data.rar’). Instead, the mass spectra raw data are in .cdf files that I cannot open despite considerable effort.
Author Response
The abstract does not adequately cover the work. For instance, the experimental approach via flux studies and siRNA is not mentioned. The sentence ‘Its production in excess and its shuttling to the mitochondrion (using simultaneous FAS and FAO) could be a mechanism that allows the cell to respond immediately to sudden increases of oxidative stress.’ is a very big leap for the reader.
The abstract was modified in the following way in order to correct the misunderstandings pointed out by the reviewer:
“Both, cytosolic fatty acid synthesis (FAS) and mitochondrial fatty acid oxidation (FAO), have been shown to play a role in the survival and proliferation of cancer cells. This study aimed to confirm experimentally if FAS and FAO coexist in breast cancer cells (BCC). By feeding cells with 13C labeled glutamine and measuring labeling patterns of TCA intermediates, it was possible to show that part of the cytosolic acetyl-CoA used in lipid synthesis is also fed back into the mitochondrion via fatty acid degradation. This results in the transfer of reductive potential from the cytosol (in the form of NADPH) to the mitochondrion (in the form of NADH and FADH2). The hypothesized mechanism was further confirmed by blocking FAS and FAO with siRNAs. Exposure to staurosporine (which induces ROS production) resulted in the disruption of simultaneous FAS and FAO, which could be explained by NADPH depletion.”
I have serious concerns regarding the UPLC/MS experiments. A triplequad delivers low resolution data and use a ‘scanning’ mode of operation, the number of spectra per second is low. The authors combine this ‘slow’ and low resolution MS instrument with very, very fast gradient LC (2 min). I expect a lot of coelution of nominally isobaric compounds. Currently, there is no way readers can assess data quality. I, therefore, suggest that the authors present overlays of extracted ion chromatograms of all isotopomers of compounds they identified. This way, the reader can be reassured that the chromatographic quality of the peaks, and the overlap of isotopomers) is sufficient to support the digested data and the conclusions drawn.
In order to address the reviewer´s concerns, we have added a figure to the supplementary material (supplementary figure S1) showing typical mass spectra and ion chromatograms of citrate isotopomers for the 3 diffetent cell lines tested in this study. The mass spectra show clearly separated isotopomers.
The following legend has been added:
Supplementary Fig. S1 Ms spectra (a,b,c) and extracted ion chromatograms (a’,b’,c’) of citrate M0-M6 isotopomers in BCC (a,a’), BT-474 (b,b’) and MCF7 (c,c’) cells. Citrate retention time 0.7-0.75 min.
Regarding the main conclusions of the study, in BCC cells it is clearly seen that the M6 (citrate with 6 labeled carbons)- 197 peak (top panel in a’, black chromatogram) is size order 1e5 order and the peak is clearly visible on the “subcromatogram”, but in bt474 (b’) and mcf 7 (c’) these values are of size order 1e4. This is the same information shown in the bar-graphs of figure 1 B.
I cannot open the cdf files in the supplementary data, in an effort to assess the LCMS data quality. I have tried to import data to XCMS, mzMine, Toppview KNIME and SEEMS but all viewers produce errors. Please convert the data to something more readable such as mzML.
The LCMS data have now been made available in mzML format as requested. The new files are included in the raw_data folder.
The use of staurosporine to induce ROS formation, seems unfortunate. Staurosporine is a very aspecific but potent inhibitor of a wide variety of protein kinases. Indeed, it also induces apoptosis which is linked to ROS production, but I find that the claim that staurosporine effects are due to ROS formation lacks evidence. I appreciate the measurement of mitochondrial depolarization, but this doesn’t preclude that the observed effects are due to other changes induced by staurosporine.
We agree with the reviewer on the non-specificity of staurosporine inhibiting protein kinases. However its ROS producing effect is a well-established fact and the fact that antioxidant agents are able to inhibit its apoptotic effects seem to indicate that staurosporine induces apoptosis by increasing ROS (Neurochem Int. 2009 55(7):581-92; Experimental Neurology 2001 171: 84-97; Journal of Biological Chemistry 2004 279, 50499-50504).
Our experiment aimed to test if NADPH depletion caused by a ROS increase (NADPH is used to reduce oxidant species in a mechanism coupled to glutathione reductase and glutathione peroxidase) would result in a decrease of the fraction of M6 citrate isotopomer, which was clearly observed in our results.
Even if staurosporine is reported to have other effects, neither alternative ways in which it could induce NADPH depletion nor possible direct inhibition mechanisms of fatty acid synthesis or beta oxidation, therefore, our initial hypothesis remains the most plausible explanation for the observed results
Round 2
Reviewer 2 Report
I have no further comments for the authors.